# Eco-Friendly Synthesis of PEtOz-PA: A Promising Polymer for the Formulation of Curcumin-Loaded Micelles

**DOI:** 10.3390/molecules27123788

**Published:** 2022-06-12

**Authors:** Monica Nardi, Emiliana Sarubbi, Satyanarayana Somavarapu

**Affiliations:** 1Department of Pharmaceutics, UCL School of Pharmacy, London WC1N 1AX, UK; sarubbiemiliana@gmail.com (E.S.); s.somavarapu@ucl.ac.uk (S.S.); 2Dipartimento di Scienze della Salute, Università Magna Græcia, Viale Europa, 88100 Germaneto, Italy

**Keywords:** green solvent, curcumin, natural antioxidants, PEtOz, Er(OTf)_3_

## Abstract

The need to develop alternative methods or to use “green” solvents constitutes an essential strategy under the emerging field of green chemistry, particularly in the development of new synthetic strategies in the field of pharmaceutic industry. We report an eco-friendly method of synthesis of poly(2-ethyl-2-oxazoline)-palmitoylate (PEtOz-PA) using Er(OTf)_3_ as Lewis’s acid catalyst in 2-MeTHF. The novel biomolecule derivative was characterized to confirm palmitoyl group substitution and employed for the formulation, characterization, and antioxidant activity evaluation of curcumin-loaded polymeric micelles.

## 1. Introduction

Consumption of polyunsaturated fatty acids (PUFA) of the omega-3 series exerts a strong positive influence on decreasing the risk factors associated with the occurrence of several degenerative diseases including cancer, cardiovascular diseases, and other inflammatory conditions [1]. These studies provide the rationale for using DHA supplements to enrich the diet of pregnant and lactating women.

## 2. Results

Biodegradable and biocompatible polymers are one of the most important platforms for biomaterials development and polymer therapeutics. In general, water-soluble non-ionic polymers are an important class of macromolecules useful for drug, gene, protein delivery, and supramolecular systems such as micelles [1] and liposomes [2].

Poly(2-ethyl-2-oxazoline) (PetOz) is a long-chain polymer biocompatible with low toxicity, good hydrophilicity, and biocompatibility, which like all the poly(2-oxazoline)s [3], are a valid alternative to standard polymers such as poly(ethylene glycol) (PEG) [4,5].

Micelles consisting of poly(2-oxazoline)s copolymers have been extensively studied due to their potential for a variety of applications including pharmaceutical applications [5], catalysis [6], and drug delivery such as nanoparticles [7].

This diversity of applications is attributed to the micelle properties that can be easily modified by designing polymer blocks [8,9] and the side chain of POXs block copolymer obtained by the polymerization method [10,11,12,13,14,15,16,17,18,19,20].

In the last years, the environmental issues that are linked to the chemical and other associated industries, especially the pharmaceutical industry, have become increasingly pertinent in the scientific community. The attention has been focused on the use of cheap, renewable, and environmentally safe starting materials such as, for example, biobased materials [21].

According to our experience with the poor toxic Er (III) salts as catalyst in mild, non-dry reaction conditions, both in homogeneous [22,23,24,25,26,27] and in heterogeneous phase [28,29,30], we planned to test the catalytic amounts Er(OTf)_3_ in green solvent [31,32,33,34,35,36] to afford the esterification of PetOz.

The use of green solvent and renewable feedstocks occupy a strategic place within the framework of green chemistry [37].

The 2-MeTHF is derived from renewable resources such as corncobs and bagasse, often giving higher reaction yields and simplified work-up summarized in three key factors: easy product isolation (2-MeTHF is not water-miscible); simple solvent recovery and drying (2-MeTHF forms a water-rich azeotrope at atmospheric pressure); and improved extraction yields (2-MeTHF reduces costs by reducing the number of extraction steps) [38,39].

In the present project, a novel hydrophobic PetOz derivative, PetOz palmitoyl (PetOz-PA), with improved organic solubility was synthesized using Er(III) triflate in 2-MeTHF. The new polymeric derivatives are characterized and employed for the preparation of curcumin-loaded polymeric micelles formulations.

Curcumin (1,7-bis-(4-hydroxy-3-methoxyphenyl)-hepta-1,6-diene-3,5-dione) is a naturally derived substance extracted from the rhizome of turmeric (*Curcuma longa*), used as a natural food coloring agent [40]. It is a potent anticancer, anti-inflammatory [41], and antioxidant polyphenol [42] but its poor aqueous solubility and rapid degradation hinder clinical application.

In this study, curcumin-loaded polymeric micelles are prepared using different amounts of PetOz-Palmitoyl (10, 20 and 40 mg) and 2 mg of curcumin (C).

## 3. Results and Discussion

### 3.1. Synthesis and Characterization of PEtOz-PA

Protecting schemes are of crucial importance in synthetic organic chemistry and represent important tools for industrial bioengineering. One of the most employed techniques for the protection of hydroxyl (OH) groups is esterification and many of the proposed methods involve the use of non-environmentally friendly or expensive reagents, toxic solvents, and involve the iterative blocking and de-blocking of all other hypothetically reactive OH in the molecule.

In this work, esterification reaction of PetOz with palmitoyl chloride was conducted to obtain a compound lipophilic enough to be used as a lipid matrix in the formulation of polymeric micelles carrying a potent antioxidant compound as curcumin. The PetOz-Palmitoyl synthesis was conducted in green solvent, 2-MeTHF, at room temperature obtaining the reaction product (Figure 1) in only 1 h without using toxic reagents. The resulting crude product was purified by flash chromatography on silica gel with a mixture of petroleum ether and ethyl acetate as eluent to afford desiderated compound (90%) as gray solid.

The product formation was confirmed by ^1^H-NMR ^13^C-NMR and FT-IR spectroscopy. Figure 1 reports the proton signals of the product. The chemical shifts shown at 0.89 ppm in the ^1^H-NMR spectrum represent protons from CH_3_ of palmitoyl group, whereas the peaks at 1.11 ppm correspond to protons of CH_3_ of PETOz group. The CH_2_ protons of palmitoyl group show the signals at 1.16 and 1.24 ppm, CH_2_ protons (α position to the carbonyl) of PETOz group at 232 ppm, CH_3_ protons of PETOz group and CH_2_ protons (α position to the carbonyl) palmitoyl group at 2.42 ppm, and CH_2_ protons (N-CH_2_CH_2_-O) of PETOz group at 3.48 ppm.

Samples of PEtOz, palmitoyl chloride, and PEtOz-PA were analyzed using FTIR to confirm the presence of the palmitoyl group in the PEtOz chain (Figure 2). The C=O stretching band of palmitoyl chloride (blue spectrum) appears at 1801 cm^−1^. After the formation of PETOz-PA, the peak shift to a lower frequency (1743 cm^−1^, orange spectrum) indicating an ester bond was formed.

### 3.2. Particle Size, Zeta Potential, and Stability of PetOzPA (C) (1:20; 1:10, and 1:5)

An experimental design was employed to find the best model that relates the ratio of PetOz palmitoyl (PetOz-PA) and curcumin (C) to mean particle size and PDI (a measure of the width of the size distribution). The hydrodynamic diameter and zeta potential were detected by DLS. As shown in Table 1, the particle sizes of PetOz-PA(C) (1:20; 1:10, and 1:5) were 200.1 ± 1.35 nm, 171.7 ± 1.01 nm, and 157.5 ± 1.328 nm. Using a greater quantity of PetOzPA increased the particle size of the formed micelles. There was no significant difference (*p* > 0.05) in PDI for the preparations, with mean PDI < 0.2, indicating that all had narrow size distribution. The PDIs of the micelles were all less than 0.2, indicating that the prepared micelles had excellent dispersion. The surface zeta potentials were −3.79 ± 0.36 mV, −5.31 ± 3.6 mV, and −13 ± 5.2 mV, indicating excellent physical stability and less systemic toxicity. Table 1 shows that the micelles were stable at 4 °C over a period of 60 days, unchanging mean particle size, and PDI at day 1, 30, and 60 (*p* < 0.05). When PetOz-PA(C) (1:20; 1:10, and 1:5) were stored at 25 °C, the mean particle size increased 2-fold compared to samples stored at 4–8 °C. This might be due to the changes in PetOz-PA behavior at different temperatures [43].

### 3.3. Entrapment Efficiencies, Drug Loading, and In Vitro Release of Curcumin

Table 2 shows the EE and drug DL of PetOz-PA(C) polymeric micelles.

The UV spectrophotometric method was established to determine the content of curcumin (C) in the formulations. The C concentration was used as an abscissa and the absorbance was used as the ordinate for standard curve construction (see Appendix A).

As shown in Table 2, different ratios of curcumin to PetOz-PA were examined. As the ratio of PetOzPA increased, the drug loading increased. Nevertheless, the encapsulation efficiency increased first and then decreased, being highest at the ratio of 1:10 (EE (%) = 96.6 ± 0.15). The drug loading of PetOz-PA(C) (1:10) was 4.6% ± 0. Hence, the ratio of 1:10 (C: PetOz-PA) was designated for future follow-up investigation.

Figure 3 shows in vitro curcumin release curves. Free curcumin was released rapidly, while the same drug encapsulated in PetOz-PA tended to be released slowly, showing better sustained-release properties. Particularly interesting are the results obtained for the curcumin-loaded polymeric micelles (PetOz-PA:C 10:1). The excellent sustained-release behavior of curcumin at a ratio of 1:10 might increase the bioavailability of curcumin in vivo.

### 3.4. Critical Micelle Concentration (CMC)

Pyrene was used as a fluorescent probe to determine the CMC of PetOz-PA(C) (1:20), (1:10), and (1:5) polymeric micelles. The CMC values were 4.27 × 10^−6^ M, 4.85 × 10^−6^ M, and 1.05 × 10^−6^ M, respectively, indicating that micelles, particularly C: PetOz-PA (1:10), had high stability and ability to maintain their structure under dilution conditions.

### 3.5. DPPH Scavenging Activity

The antioxidant activity of encapsulated curcumin was analyzed by DPPH scavenging activity assay. DPPH scavenging activity of different curcumin-loaded polymeric micelles were presented in Table 3.

Before the analysis, we measured DPPH scavenging activity of PetOz-PA, noting that the DPPH scavenging activity was low and irrelevant compared with curcumin. Therefore, we theorized that the DPPH scavenging activity of curcumin-loaded polymeric micelles came from curcumin. An increase in curcumin composition also decreased the DPPH scavenging activity.

## 4. Materials and Methods

### 4.1. Materials

Palmitoyl chloride was purchased from Tokyo Chemical Industry (UK). Poly(2-ethyl-2-oxazoline) (PetOz) average M_n_ 10,000 and PDI ≤ 1.5, 2-Methyltetrahydrofuran (Me-THF), curcumin (C, ≥94% curcuminoid content), pyrene (≥99%), and erbium(III) trifluoromethanesulfonate were obtained from Sigma-Aldrich (UK). Deuterochloroform (99.8 atom %D) was purchased from Cambridge Isotope Lab. Inc. (USA). Acetonitrile, water (HPLC grade), and methanol were purchased from Fisher Scientific (United Kingdom). Deionized water was prepared in-house (PURELAB, ELGA, UK).

### 4.2. Synthesis and Characterization of Poly(2-ethyl-2-oxazoline) Palmitoyl ester (PEtOz-PA)

To a solution of Poly(2-ethyl-2-oxazoline) (10,000 Mn; 0.1 mmol) were added a solution of palmitoyl chloride (0.15 mmol) in 2-MeTHF (8 mL) and 10 mol% of Er(OTf)_3_ under stirring. The mixture was reacted for 4 h at reflux and was evaporated under vacuum. The mixture was poured in water saturated with NaHCO_3_ and extracted with 2-MeTHF (3 × 10 mL). The organic layers were combined and dried on Na_2_SO_4_ and filtered, and the solvent was evaporated under vacuum. The purity of the product obtained was proven by Resonance Magnetic Spectrum using a NMR Bruker Avance 500 spectrometer (Bruker Instruments, USA). Samples of PEtOz, palmitoyl chloride, and PEtOz-PA were analyzed using FTIR to confirm the presence of the palmitoyl group in the PEtOz chain. Samples were analyzed with 4 scans from 500 to 4000 cm^−1^ using Perkin Elmer Spectrum100 FTIR (PerkinElmer, UK). The spectrum of PEtOz-PA was then compared to those of pure PEtOz and palmitoyl chloride. Poly(2-ethyl-2-oxazoline) palmitoyl ester: Gray solid obtained in 90% yield; ^1^H-NMR (500 MHz, CDCl_3_): δ = 3.50–3.40 (m, 484 H, CH_2_ alk), 2.45–2.40 (m, 134H, CH_2_ alk), 2.20–2.30 (m, 102H, CH_2_ alk, 2H, CH_2_ palmitoyl), 1.60–1.80 (m, 42H, CH_2_ alk), 1.25–1.35 (m, 10H, 10H, CH_2_ palmitoyl), 1.20–1.25 (m, 16H, CH_2_ palmitoyl), 1.10–1.20 (m, 343H, CH_2_alk), 0.85–0.95 (m, 3H, CH_3_ palmitoyl). ^13^C-NMR (100 MHz, CDCl_3_): 9.40, 15.29, 26.00, 29.00, 34.06, 43.21, 45.32, 66.10, 173.9, 174.2, 179.5. FT-IR (1743 cm ^1^ C=O stretching vibration).

### 4.3. Preparation of PEtOz-PA and PEtOz-PA(C) Polymeric Micelles

PetOz-PA(C) polymeric micelles were prepared using the double emulsion solvent evaporation technique as previously described [44]. The drug (C, 2 mg) and PEtOz-PA (10, 20 and 40 mg) were dissolved in CH_2_Cl_2_ (6 mL) and the mixture sonicated using a VWR ultrasonic cleaner bath USC300T (VWR International Limited, UK) until complete dissolution. The solvent was then evaporated at 200 rpm and 80 °C under vacuum until a thin film was obtained using a rotary evaporator (Hei-VAP Advantage Rotary Evaporator, Heidolph, Germany). The resultant thin film was hydrated with 10 mL of warmed distilled water and sonicated for a further 15 min until the film was fully removed and dispersed in the water. The solution was filtered through a sterile 0.22 µm filter (30 mm HPLC Syringe Filter Glass Fiber Prefilter/PTFE) and was stored at 4–8 °C before being characterized. A proportion of the samples were lyophilized using a Virtis AdVantage 2.0 BenchTop freezedryer (SP Industries, Suffolk, UK) for further analysis.

### 4.4. Physicochemical Characterization of Curcumin-Loaded Polymeric Micelles

The size and size distribution (expressed as mean hydrodynamic diameter and polydispersity index, PDI) was measured using dynamic light scattering (DLS) and laser Doppler velocimetry (LDV). Surface charge (zeta potential) of curcumin-loaded polymeric micelles were determined by Zetasizer, Nanoseries Instrument (Nano 25, Malvern Instruments, Worcestershire, UK).

Formulations were prepared with deionized water (pH 5.6) and used without further dilution. A measure of 1 mL of the sample was pipetted directly into the zeta potential DTS1070 folded capillary cell (Malvern, Worcestershire, UK) without dilution. Measurements were performed 3 times, and mean values were taken. Zeta potential was calculated from electrophoretic mobility using the Helmholtz–Smoluchowski equation by the Malvern data analysis software.

The loading efficiency of obtained formulations was determined spectroscopically and subsequently results confirmed using HPLC. Spectroscopic determination of curcumin loading was achieved by diluting of curcumin at different concentrations in dimethyl sulfoxide (DMSO). The samples were analyzed at an absorbance wavelength of 435 nm. The concentration of curcumin in each formulation was then determined using the molar extinction coefficient of curcumin determined by constructing a standard curve measuring the absorbance of known curcumin concentrations (See Appendix A). Encapsulation efficiency (EE) and drug loading (DL) were calculated using the following equations.
EE (%) = ([C]_E_/[C]_S_) × 100(1)
where
[C]_E_ is the concentration of curcumin detected spectroscopically within the nanocarriers after 0.22 µm filtration to remove unencapsulated material, and[C]_S_ is the concentration of curcumin originally added to the formulation.
DL(%) = (W_1_/W_0_) × 100(2)
where
W_1_ = total weight of entrapped curcumin in PetOz-PA.W_0_ = total weight of solid lipid, surfactant and curcumin in PetOz-PA.

Results were validated using the HPLC technique. The formulations (1 mL) were diluted to 5 mL with methanol and, to evaluate curcumin content, were analyzed by reverse-phase HPLC on a Jasco LC-NetII/ADC, UV-2075 detector. A Luna Altech 4.6 × 150 mm Adsorbosphere C18 column with 5 μm particles of bonded silica gel with a guard column (4.6 × 7.4 mm Adsobosphere C18) was used. Absorbance chromatograms were obtained at 255 nm and all the measurements were performed at room temperature.

A binary mixture of acetonitrile/water (70/30, *v*/*v*) was applied as mobile phase with a flow rate of 1 mL/min and 20 µL injection loop. The method was validated for linearity, precision, and recovery using standard solution (200 µg/mL; 140 µg/mL; 100 µg/m; 70 µg/mL; 50 µg/mL; 35 µg/mL; 25 µg/mL). The calibration curve showed good linear regression (R^2^ = 0.981) over the wide test ranges. The calibration curve was linear in the used solvent. The experiments were repeated three times

### 4.5. In Vitro Release of Curcumin

The curcumin release was quantified in vitro by a membrane dialysis (MD) method [45], a widely used and versatile method for testing in vitro drug release of particulate formulations. The end-sealed dialysis bag (MW 4 kDa) was placed in 30% ethanol in PBS (phosphate buffered saline, 0.1 M, pH 7.4, 50 mL) with continuous magnetic stirring at 100 rpm. The buffer solution was used as dissolution medium at 37 °C, and medium was collected at a determined time point. The amount of drug released in the buffer solution was monitored by UV–Vis spectroscopy, and the experiments were repeated three times.

### 4.6. Critical Micelle Concentration

The critical micelle concentration (CMC), an important parameter reflecting the formation concentration and stability of micelles, was determined using a fluorescence spectrometer (Perkin Elmer precisely LS55 luminescence spectrometer, Wellesley, USA) with pyrene as the fluorescence probe.

The concentration of blank micelles, obtained by serial dilution with pyrene using 0.2 mM pyrene dissolved in methanol at a concentration of 6 × 10^−6^ M (samples were serially diluted added into 1 mL surfactant sample, sonicated for 30 min, and incubated for 1 h at 40 °C, mixtures were kept overnight in dark condition) and the fluorescent intensity for each concentration measured at an emission from 350 to 450 nm and an excitation of 334 nm. The pyrene fluorescence of formulations was measured by fluorescence spectrometer (Perkin Elmer precisely LS55 luminescence spectrometer, Wellesley, USA) with pyrene as the fluorescence probe. The excitation/emission slits were set as 3/3 nm. At the CMC, where the amphiphilic nanocarriers begin to form, the pyrene partitions preferentially towards their hydrophobic core, causing an increase in the fluorescence intensity. This change presents as a large change in the first and third highest emission peaks (I_1_ ~372 nm, and I_3_ ~383 nm) of pyrene’s emission spectra. The CMC value was taken at the point of intersection from the two tangents drawn at high and low concentrations from the plot of the intensity ratio (I_1_/I_3_) against the log of the micellar concentration [46].

### 4.7. DPPH Assay

The DPPH radical-scavenging activity of the samples was assessed by the Gülçın method [47]. A 1,1-diphenyl-2-picrylhydrazyl radical solution was prepared by dissolving 0.0039 g of DPPH in MeOH to generate a 0.1 mM DPPH stock solution. Working solutions were prepared fresh daily by diluting the stock solution with solvent sufficiently to reduce the absorbance at 517 nm to 1.00 (±0.02). The DPPH radical solution (1 mL) was added to a solution (1 mL) of the compound to be tested in MeOH at different concentrations (see Appendix A). The obtained mixture was shaken and incubated for 40 min at room temperature in the dark. The sample absorbance was read at 517 nm. A Jenway 6850 spectrophotometer was used to follow the decrease in the absorbance of the resulting solutions. All the measurements were made in triplicate. MeOH (2 mL) and the DPPH radical solution (2 mL), used as negative controls (blank). The ability to scavenge the DPPH radical was expressed as percentage inhibition and calculated using the following equation:

The results DPPH scavenging activity (%) =
[(A_0_ − A_1_)/A_0_] × 100(3)
where A_0_ is the absorbance of the control and A_1_ is the absorbance of the compounds at different concentrations. The radical scavenger activity was expressed in terms of the amount of antioxidants necessary to decrease the initial DPPH absorbance by 50% (IC50). The IC50 value for each sample was determined graphically by plotting the percentage disappearance of DPPH as a function of the sample concentration.

### 4.8. Statistical Analysis

The results are expressed by mean ± S.E.M. from at least three independent experiments. For statistical comparisons, quantitative data were analyzed by one-way analysis of variance (ANOVA) followed by Tukey test according to the statistical program SigmaStat1 (Jandel Scientific, Chicago, IL, USA). A *p*-value less than 0.05 was regarded as significant.

## 5. Conclusions

PEtOz-palmitate was successfully prepared, as confirmed by ^1^H-NMR and ^13^C-NMR spectrum using a green and alternative procedure. The reaction was conduced in a green solvent, namely 2-MeTHF and erbium triflate catalytic conditions. PetOz-PA was obtained in excellent yields at room temperature in one hour. The resulting polymer thus obtained was used as a carrier for the preparation of curcumin-loaded polymeric micelles. The formulations prepared were made using a different ratio of curcumin to polymer derivative. PetOz-PA (C) showed excellent chemical-physical parameters and good entrapment efficiency. Particularly interesting is the result obtained for PetOz-PA(C) (1:10) with an EE value (%) equal to 97.8 ± 0.68%, much higher than 83.6 ± 0.44% and 68.5 ± 0.21% of PetOz-PA(C) (1:20) and PetOz-PA(C) (1:5) polymeric micelles, respectively. The drug loading % of PetOz-PA(C) (1:10) was 4.6 ± 0.04. By adding PetOz-PA to the formulation of curcumin-loaded polymeric micelles, the mean hydrodynamic diameter and PDI were increased, and the micelles had a high positive charge. PetOz-PA(C) with 10% C showed good stability after 60 days. The release profile for the different PetOz-PA(C) indicated slow release over the first 12 h. Excellent results were obtained in the determination of antioxidant activity using the DPPH assay. As a conclusion, the green procedure for synthesis the PetOz palmitic acid derivative (PetOz-PA) represent a promising strategy for the use of the same polymer to formulate curcumin-loaded polymeric micelles.

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
