# Peer review of "Eco-Friendly Synthesis of PEtOz-PA: A Promising Polymer for the Formulation of Curcumin-Loaded Micelles"

_molecules, 2022, doi:10.3390/molecules27123788_

Round 1

Reviewer 1 Report

The "Eco-friendly synthesis of PEtOz-PA: a promising polymer for the formulation of curcumin-loaded micelles" describes the synthesis of polymeric derivatives PEtOz palmitoyl that was used to obtain curcumin-loaded polymeric-micelles. The subject might be interesting and useful for the scientific community. But, I have a few suggestions regarding the manuscript:

- a poor characterization of the polymer is observed: what is the mass of the polymer obtained?

- it would be good to make an FT-IR spectrum for the polymer;

- 1H-NMR and 13C-NMR figures resolution is poor;

How the authors explain the fact that a low value of entrapment efficiency gives the highest value for drug loading?

The paper can be accepted for publication in the Molecules journal after minor revision.

Author Response

Dear editor, below is the answer (bold) to each question asked

 - a poor characterization of the polymer is observed: what is the mass of the polymer obtained?

The polymer is characterizated by 1H-NMR, 13C-NMR and FT-IR to demonstrate the acylation of the alcohol function. The hydroxyl group is the only functional group that can undergo esterification

- it would be good to make an FT-IR spectrum for the polymer;

Figure 2 showing FT-IR spectra was added in the manuscript.

1H-NMR and 13C-NMR figures resolution is poor;

In supplementary Material 1H-NMR and 13C-NMR figures larger and with better resolution

How the authors explain the fact that a low value of entrapment efficiency gives the highest value for drug loading?

There was an error in reporting the drug loading value for the PetOz-PA (C) sample (1: 5). The data has been appropriately corrected

Reviewer 2 Report

This paper contains a detailed account of the modification of a commercially-available polymer with a hydrophobic (long chain) ester and the subsequent use of the product for absorption and release of a drug molecule. Some advantages are observed for the hydrophobic material over the more polar precursor. 

The work involves a combination of synthesis coupled to a good deal of detailed characterisation and studies of the release of a molecule from the polymeric material. If appears to have been carried out to a high standard and the supporting information is valuable.

The paper is well-written and easy to follow, and the results will be of significant interest to readers of the journal.

My one significant suggestion for amendment - the diagram in Scheme 1 does not really show the mechanism - other arrows are needed to illustrate this. To avoid annoying organic chemists, just remove the one arrow and also the + and - charges. The reaction of alcohols with acid chlorides is very well known and we don't need a mechanism here.

Author Response

REVIEW 2

Dear editor, below is the answer (bold) to each question asked

The diagram in Scheme 1 does not really show the mechanism - other arrows are needed to illustrate this. To avoid annoying organic chemists, just remove the one arrow and also the + and - charges. The reaction of alcohols with acid chlorides is very well known and we don't need a mechanism here.

scheme 1 was suitably modified according to the suggestions provided by the review
